# Northern Hemisphere Stratospheric Temperature Response to External Forcing in Decadal Climate Simulations

Abdullah A. Fahad[1,5], Andrea Molod[1], Krzysztof Wargan[1,6], Dimitris Menemenlis[2], Patrick Heimbach[3], Atanas Trayanov[1,6], Ehud Strobach[4], and Lawrence Coy[1,6]

[1]Global Modeling and Assimilation Office, NASA Goddard Space Flight Center, 8800 Greenbelt Rd, Greenbelt, 20771, MD, USA

[2]Jet Propulsion Laboratory, California Institute of Technology, 4800 Oak Grove Dr, Pasadena, 91109, CA, USA

[3]Jackson School of Geosciences, University of Texas at Austin, 2305 Speedway Stop C1160, Austin, TX, US

[4]Agricultural Research Organization, Rishon LeTsiyon, Israel

[5]ESSIC, University of Maryland, 5825 University Research Ct suite 4001, College Park, 20740, MD, US

[6]Science Systems and Applications, Inc., 10210 Greenbelt Rd., Suite 600, Lanham, 20706, MD

**Correspondence:** Abdullah A. Fahad (a.fahad@nasa.gov)

**Abstract.** To predict the future state of the Earth system on multiyear timescales, it is crucial to understand the response to changing external radiative forcing ($CO_2$ and Ozone). Analyzing the Northern Hemisphere (NH) winter stratospheric polar vortex temperature, we found a general temperature decrease in the reanalysis data (1982–2020), the expected trend with increasing $CO_2$, except for a sharp warming during period 1992–2000. Results from 1°GEOS-MITgcm coupled general circulation model simulations of past decades show a similar increase in the NH polar stratospheric temperature during 1992–2000 and a decrease during 2000–2020. To isolate the influence of external forcing, we conducted a series of 30-year-long "perpetual" time-slice experiments in which the external forcing for a particular year is held fixed at its values for 1992, 2000, and 2020. Each simulated year of these perpetual experiments is forced with the $CO_2$, Ozone, anthropogenic aerosol emissions, and trace gases of that year, but none of the simulations include any explosive volcanic forcing. The increasing and then decreasing temperature trend is also manifest in the CMIP6 historical simulations performed with models that include a well-resolved stratosphere. The configuration of the perpetual experiments rules out a response to volcanic emissions or a change in the phase of decadal modes of variability as explanations for the warming rather than the expected cooling behavior. Analysis of the temperature budget showed (only significant terms are discussed) that the polar stratospheric temperature behavior is dictated by meridional eddy transport of heat resulting from changes in $CO_2$ and Ozone over the past decades.

## 1 Introduction

Seasonal to decadal climate prediction is a relatively new frontier in climate prediction research, and accurate prediction relies on the ability of the models to estimate the proper initial state, the proper internal variability, and the proper response to the natural and anthropogenic external forcing (Smith et al., 2007; Keenlyside et al., 2008; Pohlmann et al., 2009; Kirtman et al., 2013; Meehl et al., 2014; Marotzke et al., 2016; Santer et al., 2023). The strong reliance of forecasts at multiyear time scales on both internal variability and the response to changes in external forcing provides a particular challenge for prediction these long

lead times. The ability to understand and predict how the external forcing, such as changing concentrations of $CO_2$ and ozone-depleting substances (ODSs) in the atmosphere (affecting Ozone), drives the climate, as well as how the internal variability on multiyear timescales drives the climate, are both critical for multiyear climate prediction.

Given the established connection between the northern hemisphere (NH) surface weather and climate and the circulation and temperature of the NH lower stratosphere (Thompson et al., 2002; Waugh et al., 2017; Kolstad et al., 2010; Norton, 2003), improving the understanding of the influence of observed levels of external radiative forcing in that region will contribute to the understanding and possible improvement of seasonal to decadal climate prediction skill. For example, the lower stratosphere polar vortex in the NH winter can influence the troposphere and near-surface extreme weather and climate (Thompson et al., 2002; Waugh et al., 2017; Kolstad et al., 2010; Norton, 2003). Analyzing 51 years of reanalyses data and coupled climate models, Kolstad et al. (2010) found that the lower stratosphere polar vortex and temperature associated with it influences the cold air outbreaks in the NH high-latitude regions. Thompson et al. (2002) found a significant relationship between the polar vortex and surface extreme cold events in the NH mid-high latitudes. They concluded that a high level of prediction skill of NH surface cold events can be achieved by predicting lower stratospheric vortex circulation and temperature. The future evolution of Arctic stratospheric ozone also critically depends on the long-term behavior of lower-stratospheric temperatures under increasing concentrations of greenhouse gases and declining ODSs, although there is currently little agreement on the details of the effects of climate change on the Arctic stratospheric polar vortex and polar ozone depletion (Rex et al., 2004; Rieder and Polvani, 2013; Rieder et al., 2014; von der Gathen et al., 2021). The current study enhances understanding of multiyear climate predictability by analyzing the impacts of $CO_2$ and ozone on Northern Hemisphere climate dynamics using a General Circulation Model (GCM). It offers insights into external forcing influences on multi-year climate, which is crucial for improving future climate prediction accuracy.

Theory and previous studies have shown that with increased $CO_2$ levels in the atmosphere the global mean surface temperature increases, whereas the stratospheric temperature decreases (Manabe and Wetherald, 1967; Fels et al., 1980; Ramaswamy et al., 2001; Austin et al., 2003; Eyring et al., 2007; Randel et al., 2009; Gillett et al., 2003; Rind et al., 1992, 1998; Manzini et al., 2014; Santer et al., 2023). Using a 40-level General Circulation Model (GCM), Fels et al. (1980) conducted sensitivity experiments with increased levels of $CO_2$ and/or ozone reduction in the atmosphere and found the stratospheric temperature to decrease due to both perturbations. A uniformly doubled $CO_2$ in the atmosphere was found to reduce the temperature from the tropopause to approximately 50km uniformly over all latitudes. Rind et al. (1992, 1998) used sets of climate sensitivity experiments with doubled $CO_2$ levels and found similar results.

Using observed satellite, in-situ, and reanalyses data, Ramaswamy et al. (2001) found significant stratospheric cooling during 1960-1990, including at the NH pole. The study concluded that the observed cooling trend in the stratospheric temperature is due to the radiation change resulting from the depletion of lower stratospheric ozone and to a lesser extent from changes in well-mixed greenhouse gases. However, they also found that a large dynamical interannual variability exists in this region from the winter to the spring season. Randel et al. (2009) used updated satellite, radiosonde, and lidar observations from 1979-2007 and found a mean cooling of between 0.5 K/decade and 1.5 K/decade in the stratosphere, with the greatest cooling in the upper stratosphere. The stratospheric temperature anomalies, however, remained constant without any significant trend from the year

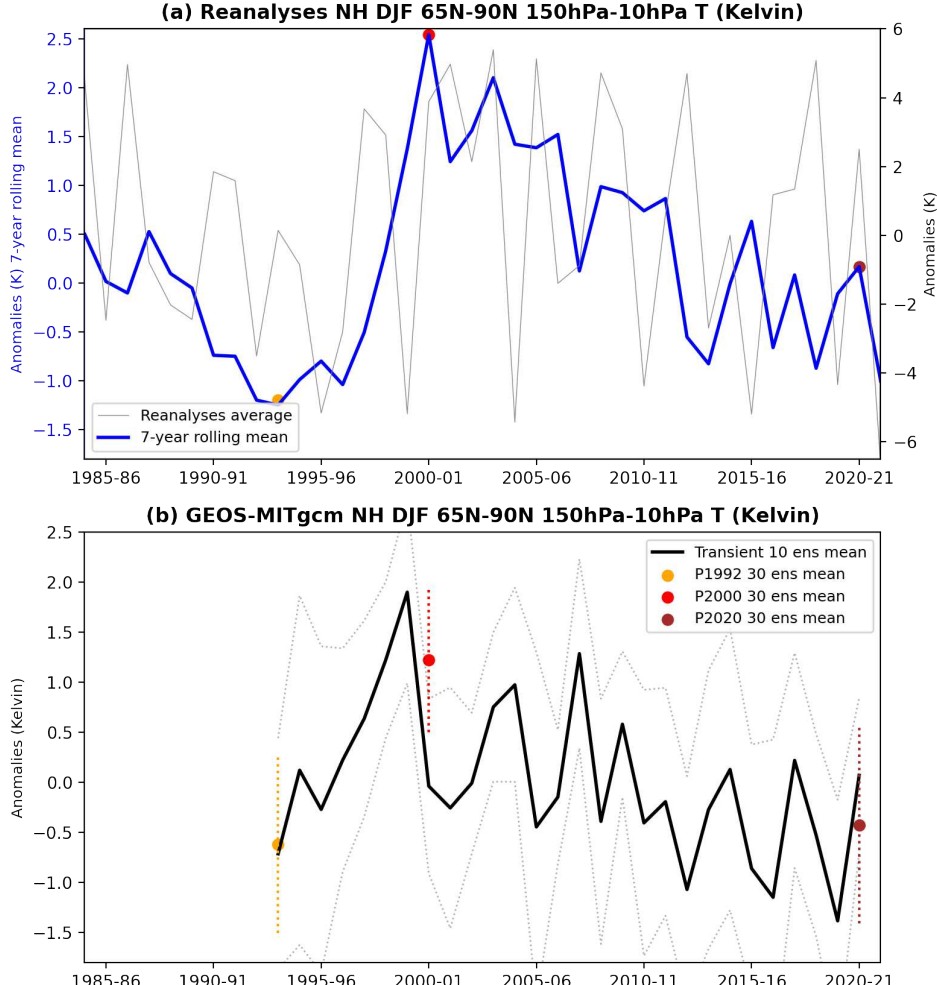

**Figure 1.** NH DJF mean 150hPa-10hPa and 65°N-90°N mean air temperature anomalies for (a) Reanalysis (MERRA2 and ERA-5), and (b) GEOS-MITgcm. The blue line in (a) shows a 7-year running mean. The GEOS-MITgcm 10-member mean transient simulation shown in (b) with bold black line with 1-std spread (grey dotted lines). The 30-member ensemble mean of perpetual experiments is shown in (b) for P1992 (orange line), P2000 (red line), and P2020 (brown line). The 1-std spread of the ensemble is shown with dotted lines. Anomalies are based on 1982-2020 for reanalysis and 1992-2020 for GEOS-MITgcm.

1995 to 2005. The study assumed that this absence of any statistical trend was due to the high level of natural (dynamical) variability that is present in the NH polar region. Randel et al. (2016) used observations from the series of Stratospheric Sounding Unit sensors to estimate linear trends in stratospheric temperatures between 1979 and 2015. They found a cooling trend, increasing with altitude between the lower the middle, and upper stratosphere (from approximately -0.1 K to -0.5 K per decade). This trend was found to be larger in the first half of the record (1979-1997) than in the following decades. These findings are broadly consistent with the results obtained by Seidel et al. (2016). Polar stratospheric temperature is closely related

to the strength of the mean stratospheric overturning circulation (the Brewer-Dobson Circulation: BDC) with the intensity of air subsidence over the high latitudes in winter being the main control knob for temperature via adiabatic heating. Climate models project an overall acceleration of the BDC with increasing greenhouse gas concentrations (Butchart, 2014; Abalos et al., 2021, and references therein). However, this simple picture is complicated by inter-hemispheric asymmetries in the BDC trends (Stiller et al., 2012, 2017; Ploeger and Garny, 2022) and, more broadly, by differential structural evolution of various aspects of the BDC (Bönisch et al., 2011; Garfinkel et al., 2017; Oberländer-Hayn et al., 2016). Additional complexity arises from the impacts of ozone recovery on the BDC (Abalos et al., 2019; Polvani et al., 2018). The latter two studies demonstrate that the impact of ozone depletion (up to approximately the year 2000) and subsequent recovery outweigh those of the gradual increase in CO2 during the same period leading to a pattern of acceleration and deceleration of the BDC over the southern polar region during the austral summer. Those studies, however, did not find a similar trend in the northern polar cap temperatures. The study of Zhou et al. (2019) demonstrates a nonlinear response of the NH polar vortex temperature to the tropical western Pacific heating associated with SST change during DJF. The study showed that increasing levels of heating over the western tropical Pacific excite stationary Rossby-type waves that propagate to the NH high-latitude upper troposphere, and impact the temperature of the polar vortex in a non-linear fashion. However, the details of the pathway by which the tropical diabatic heating forces the NH polar vortex still remain unexplored.

Analyzing reanalysis data, we found NH polar vortex temperature decreases in the satellite record (1982–2024) during winter; however, a sharp warming period exists from 1992 to 2000 (Fig. 1a). This warming trend is opposite to what we expect from a response to a $CO_2$ increase in the atmosphere. The present study aims to explore NH winter polar stratospheric temperature change in recent decades using GEOS-MITgcm coupled decadal climate simulations. We investigate 'perpetual' time-slice ensemble experiments with our GEOS-MITgcm coupled climate model to understand what drives the initial cooling at the end of the period (1992), the peak of sharp warming (2000), and one of the recent cooling years (2020) (Fig. 1a). Specifically, the study examines the modeling of this temperature evolution, attributes the temperature change to the contribution of external radiative forcing from $CO_2$ and ozone change, and identifies the dynamical pathway by which this temperature evolution is simulated in the model. Section 2 of this study describes the model, the experimental design, and the reanalyses data that were used in this study. The findings of this study are documented in section 3. Sections 4 and 5 are the discussion and conclusions of this study based on the results from Section 3.

## 2    Methodology and Data

Observationally based estimates were used from Modern-Era Retrospective analysis for Research and Applications, Version 2 (MERRA-2) (Gelaro et al., 2017; GMAO, 2015) and ECMWF Reanalysis 5th Generation (ERA5) (Hersbach et al., 2020) reanalyses for the years 1982-2024. Long-term temperature variability and trends in reanalyses are affected by step changes in the assimilated observations and generally have to be treated with caution. However, a detailed evaluation conducted as part of the Stratosphere-Troposphere Processes And their Role in Climate Reanalysis Intercomparison project (Fujiwara et al., 2022; Long et al., 2017) indicates that these changes affect mainly the upper stratosphere (not considered here) and occur mainly

before 1998. Post-1998 stratospheric temperatures at pressures greater than 10 hPa are robust among the modern reanalyses (Fujiwara et al., 2022).

To understand the NH stratospheric climate response in the past decades, we used the Goddard Earth Observing System-MITgcm (GEOS-MITgcm) coupled earth system model at a nominal 1-degree horizontal resolution in the atmosphere and ocean, with 72 hybrid vertical levels in the atmosphere (top lid 0.01 hPa) and 50 levels in the ocean. Details of the model can

be found in (Strobach et al., 2022), but some aspects relevant to this study are described here. The atmospheric model includes the finite volume dynamical core on a cubed sphere grid (Putman and Lin, 2007), a full suite of physical parameterizations including the two-moment cloud microphysics (which includes the aerosol indirect effect) of Barahona et al. (2014), the land model of Koster et al. (2000), and is coupled to the Goddard Chemistry Aerosol Radiation and Transport (GOCART) interactive aerosol model Chin et al. (2002); Colarco et al. (2010). The aerosol emissions in the simulations described here

do not include explosive volcanics. The MITgcm has a finite-volume dynamical core (Marshall et al., 1997) with a nonlinear free-surface and real freshwater flux (Adcroft and Campin, 2004). The MITgcm was configured with a 100 km horizontal grid on a "Lat-Lon-Cap" grid (Forget et al., 2015), and 50 vertical levels.

To investigate the NH lower stratospheric temperature change on multiyear time scales, we conducted both transient simulations and 30-year long 'perpetual year' time-slice simulations, for which the external forcing in a particular year is repeated 30

times. The 'perpetual year' simulation is similar to CMIP6 pre-industrial simulations where the long-term ($\sim$500 years) simulation is fixed to 1850 forcing for each year and only driven by internal variability (Eyring et al., 2016). We ran 3 'perpetual year' simulations; the year 1992 (P1992), the year 2000 (P2000), and the year 2020 (P2020), each forced with the respective year's $CO_2$ and Ozone ($O_3$). The specific years chosen for these simulations are the inflection points in the NH lower stratospheric temperature time series shown in Figure 1a and indicated by the filled circles. In these 'perpetual' simulations the annual cycles

of $CO_2$ emissions and ozone are fixed for the perpetual year as the simulation progresses, resulting in a repetition of the same year's external forcing 30 times. The 30 years of simulation are regarded here as a 30-member ensemble of simulations of the 'perpetual' year, as the initial states for each perpetual year are random. The 30-member ensemble mean of these experiments does not include a realistic simulation of the phase of low-frequency modes of internal variability, so the differences among the perpetual experiments are due only to the influence of the differences in external forcing. The annual global mean $CO_2$ level

for P1992 is 356ppm, for P2000 is 368ppm, and for P2020 is 413ppm. There are no explosive volcanic emissions included in any of the perpetual simulations.

The initial conditions for the 30-year long perpetual experiments are all taken from the same spun-up GEOS-MITgcm state (Year 2000), originally initialized with Modern-Era Retrospective analysis for Research and Applications, Version 2 (MERRA-2) (Gelaro et al., 2017) (MERRA-2) and Estimating the Circulation and Climate of the Ocean (ECCO) data (Wunsch and

Heimbach, 2007; Forget et al., 2015) (ECCO). We regard the 30-year long perpetual experiments as 30 member ensembles that are all forced by the corresponding year's $CO_2$, ozone and aerosol emissions (Zhou et al., 2024; Alexander et al., 2004; Portal et al., 2022). The initial conditions for the 10-member ensemble of transient experiments were randomly chosen from the 1992 perpetual experiment. Each year of the transient experiment is forced with observed $CO_2$ (CMIP, Eyring et al. (2016)), $O_3$ and aerosol (Randles et al., 2017) emissions (excluding explosive volcanics) from the correct year of the simulation. The

low-frequency SST modes are out of phase across these ensemble members, minimizing their influence when computing the ensemble mean (more details are in the Discussion section).

The GEOS-MITgcm coupled model transient simulations reproduce the mean state and variability of the polar vortex reasonably well as compared to reanalysis. The geopotential height at 10 hPa from the 10-member ensemble mean for January over 1992–2020 shows a similar mean state and location of the NH polar vortex during winter compared to reanalysis (MERRA-135 2: 1992–2020) (Fig. 2 a,b). The 30-year mean January momentum-flux variance (U'V') associated with the vortex wind jet, calculated from sub-monthly fields, further shows that the stability and variability of the vortex core are simulated reasonably well in the GEOS-MITgcm compared to the reanalysis (Fig. 2 c,d). The model, however, produces a bit weaker mean geopotential height ($\sim$ 4km higher at the core) and momentum flux variance (weaker $\sim 25\ m^2/s^{-2}$ near Greenland) compared to MERRA-2, suggesting that the simulated vortex may be a bit more resilient under extreme wave forcing. This could be a result 140 of a lack of higher vertical resolution in the stratosphere. However, the close agreement in both mean height structure and eddy flux confirms that the model faithfully represents the polar vortex's stability and natural variability.

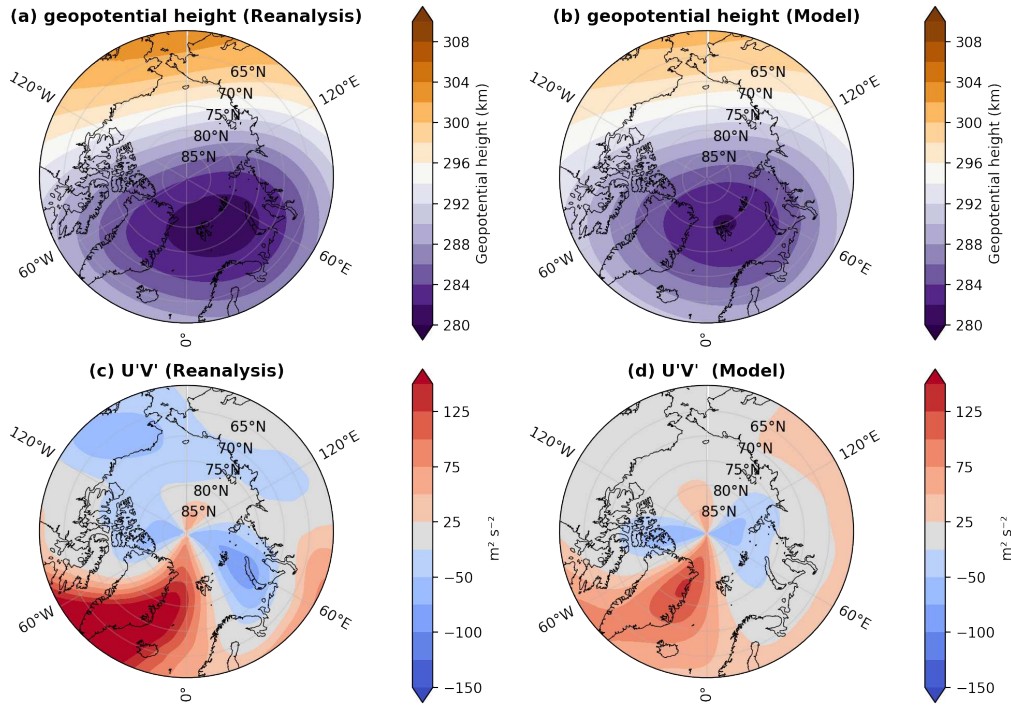

**Figure 2.** The mean state and variability of the polar vortex during January at 10 hPa are shown by the monthly mean geopotential height (km) and momentum flux ($m^2/s^{-2}$) (U'V', calculated from sub-monthly fields) for the MERRA-2 reanalysis (1992–2020) in panels (a) and (b), and for the GEOS-MITgcm 10-member ensemble mean (1992–2020) in panels (c) and (d). The model reproduces a mean state and stability similar to those in the reanalysis.

As part of the analysis of results, the diabatic heating is taken directly from the model output as a sum of temperature tendencies due to longwave radiation, shortwave radiation, moist processes, turbulent sensible heating, and gravity wave drag

(Molod et al., 2015; Bosilovich, 2015; Fahad et al., 2020). The eddy component of the meridional atmosphere heat transport is calculated following Holton and Hakim (2012) (See Appendix, section 6 for more details). We analyzed the contributions to the meridional heat transport by steady symmetric circulations ($[\overline{V}][\overline{T}]$), stationary eddies ($[\overline{V^* \ T^*}]$), and transient eddies ($[\overline{V'T'}]$, see Section 6). Here, $T$ and $V$ are the temperature and meridional components of the wind field, respectively. Square brackets denote the zonal average and overbar is the time average (DJF seasonal mean). The corresponding departures from the time and zonal averages are indicated by an asterisk and a prime, respectively.

Winter and springtime polar stratospheric temperature is highly correlated with the strength of air subsidence over the high latitudes, which is, in turn, related to the intensity of wave activity in the extratropics (Shaw and Perlwitz, 2014; Newman et al., 2001). To quantify the strength of the residual circulation we use the transformed Eulearian mean (TEM) mass stream function calculated as in Birner and Bönisch (2011). The Eliassen Palm (EP) flux is defined following Edmon Jr et al. (1980). The EP flux is a two-component vector field that measures the intensity and direction of zonally averaged wave propagation in the meridional plane. Its divergence is closely related to the strength of wave-zonal flow interactions (Andrews et al., 1987). In particular, convergence (negative divergence) of the EP flux indicates wave dissipation, which decelerates of the zonal flow and accelerates mass subsidence at high latitudes. The reverse is true for positive divergence.

To calculate the significance of the difference in means between the ensembles of different experiments, we primarily used a two-sided t-test. Additionally, we performed a nonparametric bootstrap significance test using $\alpha = 0.05$ (95% confidence) and 1,000 bootstrap samples to assess whether the mean difference between the two datasets was statistically significant without assuming normality. We found similar conclusions for all analyses with the bootstrapping significance test, with results that were slightly more stringent than those from the t-test.

## 3   Results

### 3.1   NH Stratosphere Response in Decadal Climate Simulations

Examination of DJF temperatures averaged between 150 hPa and 10 hPa and between 65° N and 90° N from MERRA-2 and ERA5 (Fig. 1a) reveals an overall negative trend over the NH polar vortex from 1982 to 2020, with the exception of a strong warming trend from 1992 to 2000. The DJF mean temperature trend during 1992–2000, although over a short timescale, is strong and becomes evident in the 7-year running-mean (the warming trend is insensitive to the window size) time series, which filters out some of the inter-annual variability (Fig. 1a). This sharp warming from 1992 to 2000 contradicts the general expectation that the stratosphere cools as the sea surface and troposphere warm in response to increased $CO_2$. It is also the only period in the reanalysis record exhibiting such a pronounced, significant temperature trend.

In this study, we focus on the years 1992 to 2020 to investigate the temperature behavior, and compare the end of the initial cooling period (1992), the peak of the transient warming (2000), and the recent resumption of cooling at the end of the time series (2020). The 10-member ensemble mean of the transient GEOS-MITgcm climate simulations from the years 1992-2000 and 2000-2020 (Fig. 1b) shows similar behavior in the NH stratospheric high latitude temperature, with a strong positive temperature trend in the first period of 1992-2000, and a strong cooling temperature trend from the year 2000-2020.

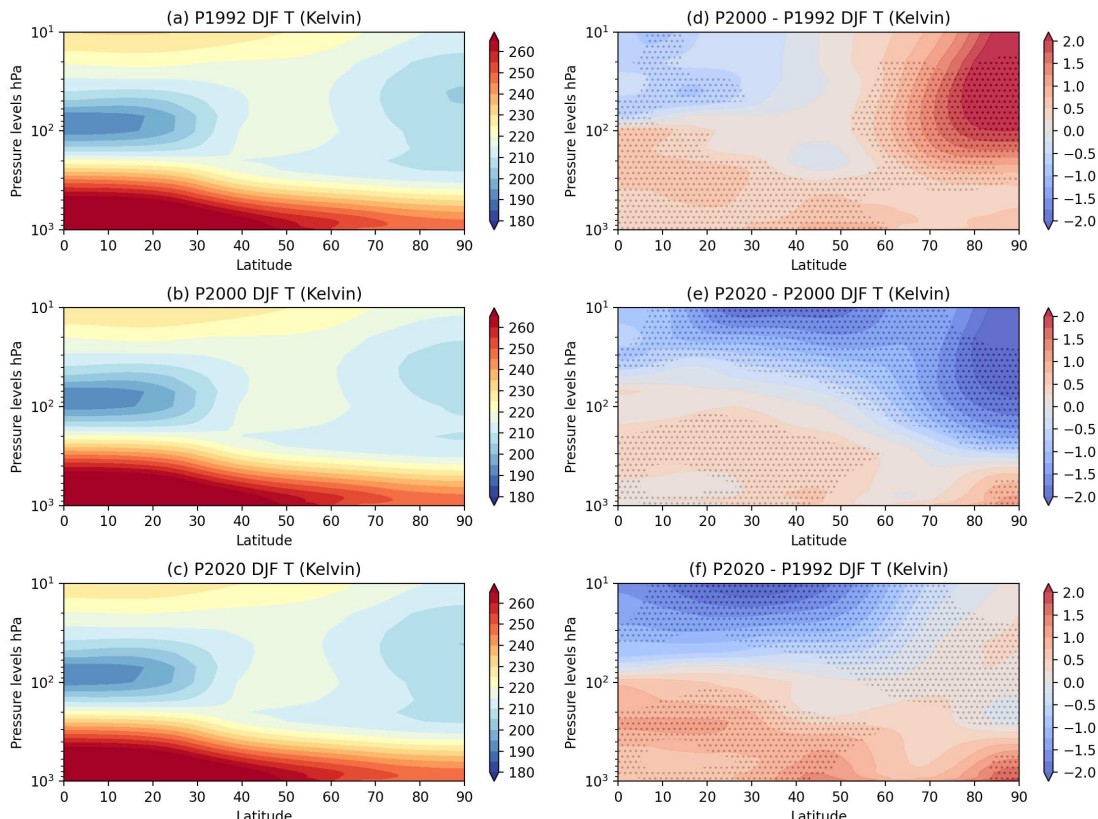

**Figure 3.** Zonal Mean NH DJF air temperature (T) mean for (a) P1992, (b) P2000, (c) P2020, and (d) difference between P2000 - P1992, (e) difference between P2020 - P2000, (f) difference between P2020 - P1992; (unit: Kelvin). Figures are stippled at 95% significance computed using a difference of means 2-sided t-test from the 30-member ensemble sample.

The polar stratospheric temperature behavior found here is consistent with the findings of Fu et al. (2019), who found a positive lower stratospheric temperature trend in the 1990s in both hemispheres winter (especially during September for the Southern Hemisphere), whereas the trend is negative during years 2000 to 2018. Focusing on the Southern Hemisphere, Fu et al. (2019) concluded that this overall temperature trend in September months in the Southern Hemisphere is most likely due to the ozone healing process after the year 2000.

In the transient experiments and reanalysis data, internal low frequency variability might influence the year to year changes, and here we wish to isolate the influence of the external forcing. The 30-member ensemble mean of the perpetual experiments will substantially minimize the influence of internal variability, and so any behavior seen in the transient experiment that is replicated in the perpetual experiments can be attributed to external forcing.

Figure 3 shows the zonal mean NH DJF mean air temperature (T) as a function of pressure for the perpetual experiments in the left panels (a, b, and c), and the difference between pairs of perpetual experiments in the right panels (d, e and f). The

difference between the 30-member ensemble mean of perpetual experiments P2000-P1992 (Figure 3d) shows that there is a strong warming in the NH stratosphere at high latitudes (65°N-90°N). In contrast, the difference between P2020 and P2000 shows a strong cooling in the NH stratosphere's high latitudes (Fig. 3e). Due to this opposing stratospheric temperature change during the two intervening periods, the difference between P2020 and P1992 shows no significant change in much of the region. The significance tests to calculate 95% confidence were conducted using t-tests and bootstrapping, and yielded similar confidence conclusions for the warming and cooling patterns. In passing, we note that the stratospheric cooling at midlatitudes is consistent with the expected radiative effects of increasing $CO_2$.

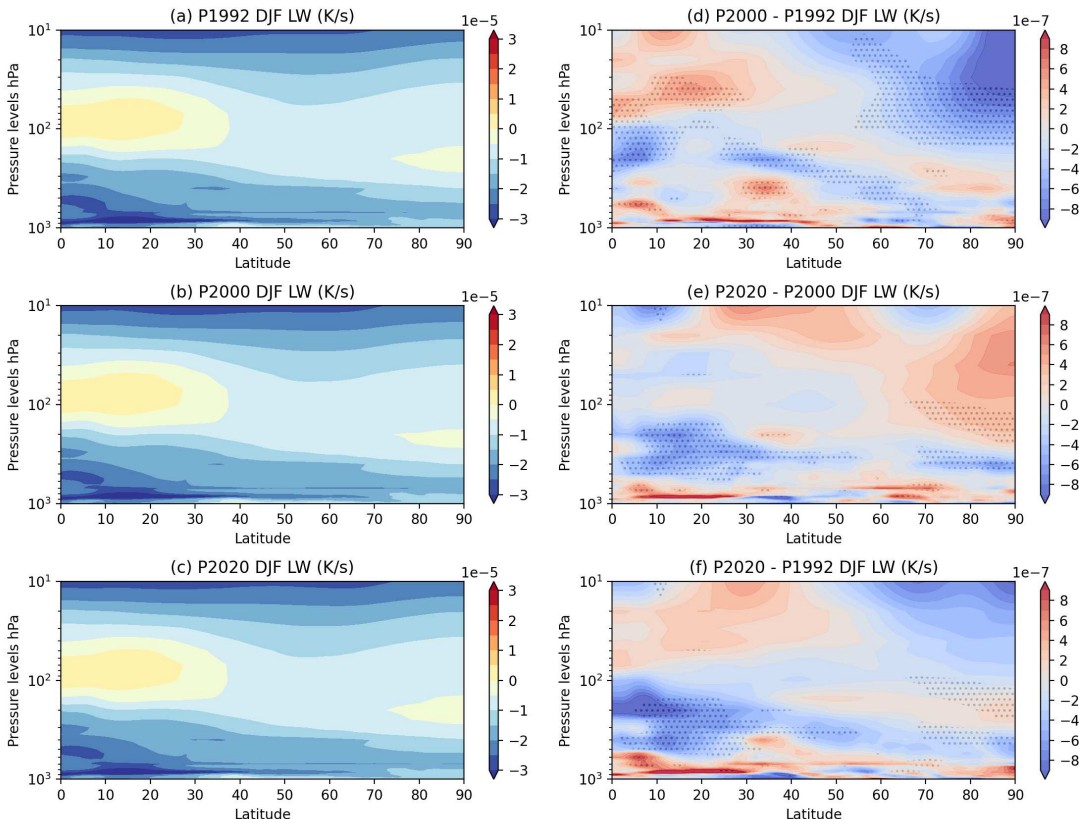

**Figure 4.** Zonal Mean NH DJF Longwave Cooling mean for (a) P1992, (b) P2000, (c) P2020, and (d) difference between P2000 - P1992, (e) difference between P2020 - P2000, (f) difference between P2020 - P1992. Figures are stippled at 95% significance computed using a difference of means 2-sided t-test from 30-member ensemble sample.

As discussed in Section 2, trends in reanalyses may be due to changes in the mix of observations used in the assimilation as well as to low-frequency internal variability, and/or to changes in the external forcing. The agreement among our perpetual experiments that used constant $CO_2$ and ODS levels from specific years, our transient experiments, and reanalyses from the

same period suggests that external radiative forcing related to the $CO_2$ and ozone concentrations in the atmosphere is the key factor responsible for the change in stratospheric temperature trends.

To analyze the proximate cause of the warming and then cooling of the high-latitude stratosphere, the individual temperature tendency terms from the different physical and dynamical model processes were examined. The largest tendency terms are those due to the longwave (there is little solar forcing in boreal winter) and those due to dynamical processes. The longwave tendency in the perpetual experiments shows a negative change in longwave cooling (cooling increases) from the P1992 to P2000, which would tend to cool the atmosphere, and a positive change (cooling decreases) from the P2000 to P2020 in the NH high latitude stratosphere (Figs. 4d & e), which would tend to warm the atmosphere. The behavior during both periods suggests that the change in the longwave cooling rates are a result of the temperature change, rather than a cause of the change. This leaves the dynamical tendency term as the cause of the temperature change.

## 3.2 Dynamical Mechanism of Forced Change

The polar cap wintertime temperature at high latitudes is controlled by the intensity of adiabatic warming from air subsidence balanced by radiative cooling. High-latitude temperature is, therefore, largely driven by the vertical component of the residual circulation in that region on interannual time scales (Newman et al., 2001). In the TEM formulation the variability of the latter is determined primarily by the zonal mean horizontal eddy heat flux (alongside the zonal mean vertical velocity) (Shaw and Perlwitz, 2014). To articulate the role of the dynamic tendency terms on the NH stratospheric polar temperature we therefore begin by analyzing the eddy heat flux from the three sets of simulations. The sum of the stationary and transient components of the heat flux are shown in Fig. 5. The meridional eddy heat flux to the NH pole increases from the P1992 to the P2000 result, whereas it decreases from the P2000 to the P2020 result(Fig. 5). The increased eddy heat flux from the P1992 to the P2000 experiment is due to both stationary wave $(V^*T^*)$ (Fig. S2d) and transient wave $(V'T')$ (Fig. S3d) activity. The meridional eddy heat flux decrease from the P2000 to P2020 is also due to both stationary wave $(V^*T^*)$ (Fig. S2e) and transient wave $(V'T')$ activity decrease (Fig. S3e). The analysis from here onward in the study makes no distinction between the contributions from transient and stationary waves.

Figure 6 shows the residual mass stream function (see equation A2) for the periods of interest and its difference from one period to the other. Evident is an intensification of the residual circulation between 1992 and 2000 (red shading in Fig. 6a), followed by a weakening after 2000 (blue shading in Fig. 6b). In particular there is an increase (decrease) of air subsidence in DJF over the high latitudes in the early (late) period. This implies a strengthening (weakening) of adiabatic warming in P2000-P1992 (P2020-P2000), and this finding is qualitatively consistent with the pattern of temperature changes in Figures 3 and 1. We note that the differences in the intensity of subsidence over the pole in Figure 6 are in agreement with the convergence of mean eddy heat flux shown in Figure 5. While the polar temperatures in 2020 approximately return to the 1992 values (Figure 1) the intensity of the streamfunction in 2020 still exceeds that in 1992 (Figure 6).

The stratospheric residual circulation is driven by momentum deposition from Rossby wave breaking interacting with the zonal flow in the midlatitudes (Andrews et al., 1987). This is quantified by the EP flux convergence, wherein positive convergence decelerates the zonal mean zonal wind and induces subsidence over high latitudes. Figure 6 (d,e,&f) shows the changes

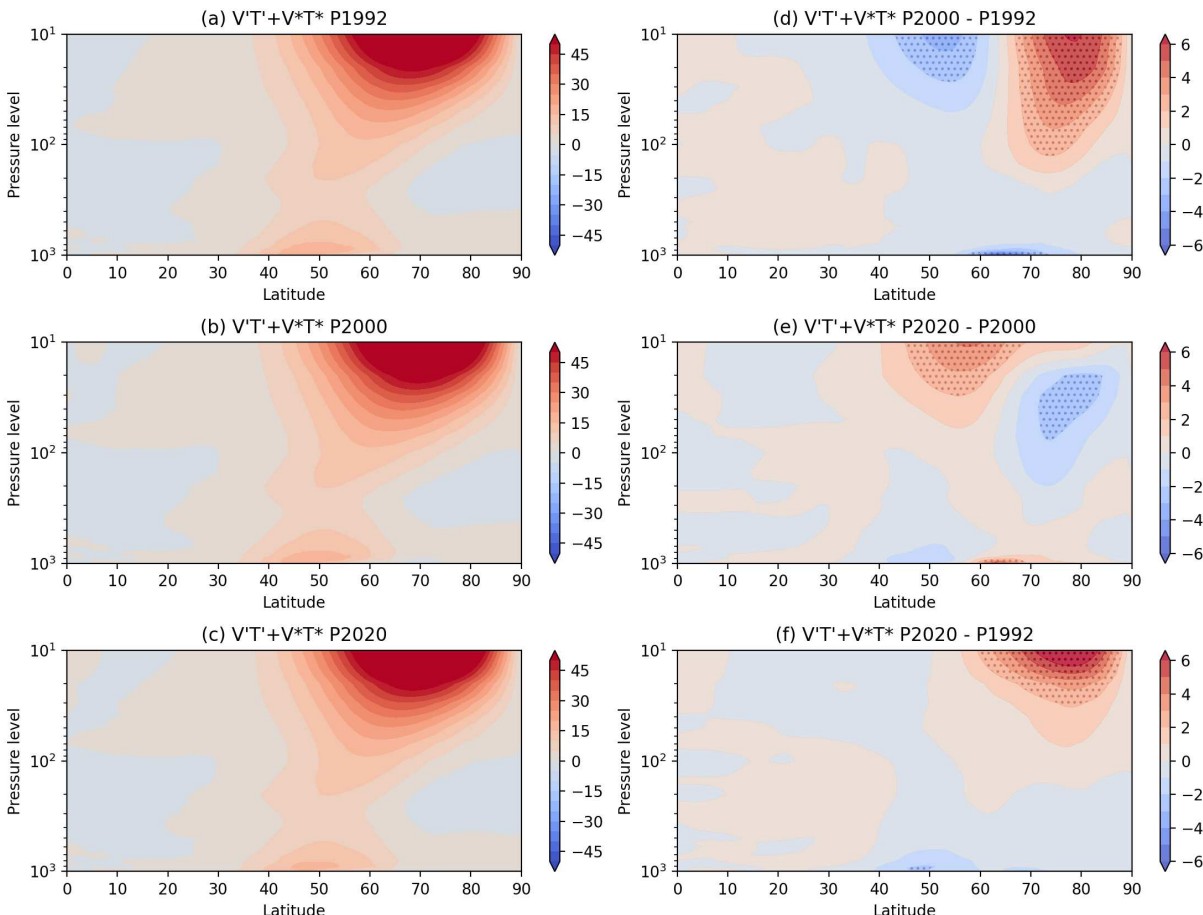

**Figure 5.** Zonal mean DJF mean V'T' + V*T* in $Kms^{-1}$ for (a) P1992, (b) P2000, (c) P2020, (d) P2000-P1992, (e) P2020-P2000, and (f) P2020-P1992. Figures are stippled at 95% significance computed using a difference of means 2-sided t-test from 30-member ensemble sample.

in the EP flux and its convergence between P2000-P1992, P2020-P2000 and P1992-P2000. The upward pointing arrows and the increase in convergence at midlatitudes in panel (d) indicate an intensification of wave activity and wave breaking between 1992 and 2000. This is consistent with the strengthening of the residual circulation during that period (Figure 6a) and, consequently, with the increase in polar temperature. The converse is true for the changes between 2000 and 2020. The overall change between 1992 and 2020 (panel f) is also consistent with the intensification of the residual circulation over the same period (Figure 6c).

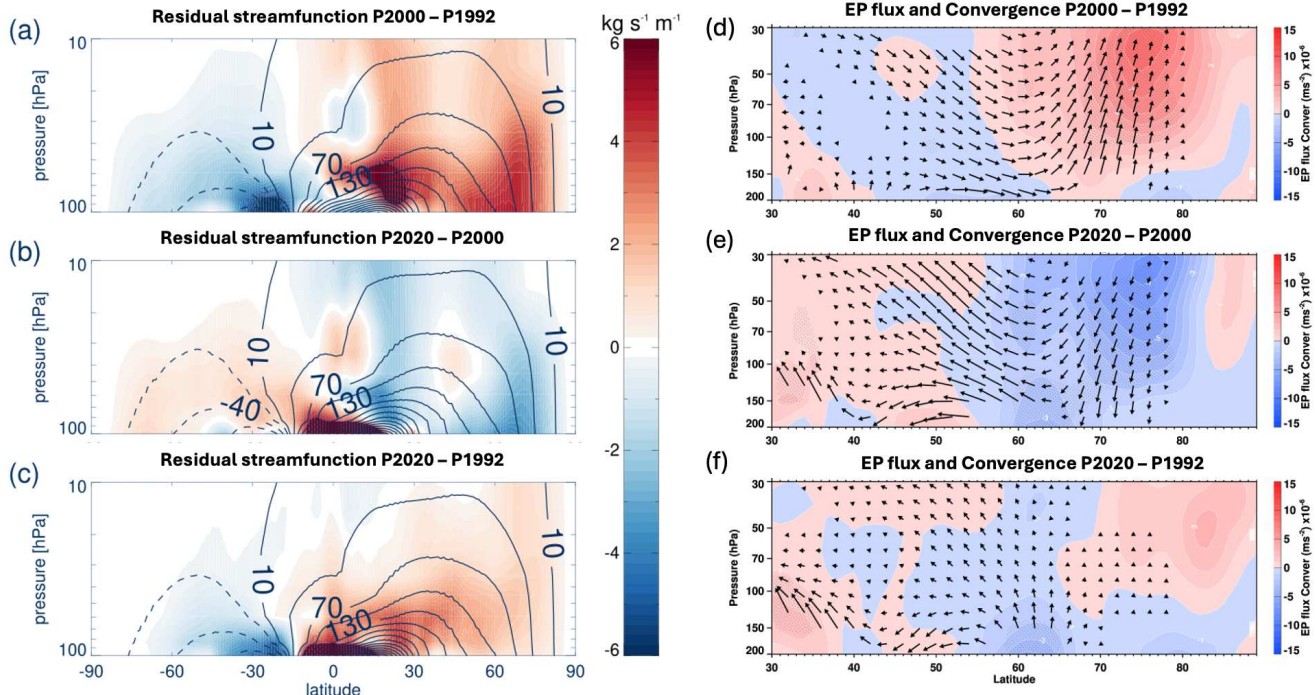

**Figure 6.** Mean Residual streamfunction (a,b,c), and EP flux with convergence (d,e,f) during DJF from model simulations. Contours: mean residual streamfunction divided by the Earth's radius for 1992 (a and c) and 2000 (b). Shading: the streamfunction change for (a) P2000-P1992, (b) P2020-P2000, and (c) P2020-P1992. EP flux (vector) (unit: $m^2/s^2$) and convergence (shaded) (unit: $m/s^2$) for (d) P2000-P1992, (e) P2020-P2000, and (f) P2020-P1992. Convergence is shown in red and divergence is shown in blue colors. Figures are stippled at 95% significance computed using a difference of means 2-sided t-test from 30-member ensemble sample.

## 4 Discussion

Our 10-member mean transient external forcing experiments, as well as our perpetual experiments, show an opposing NH lower stratosphere temperature trends for the years 1992-2000 and for the years 2000-2020, in agreement with reanalysis data. We have shown that this pattern is directly due to the strenthening and then weakening of the dynamical heating (as opposed to radiative heating), consistent with the strengthening (1992-2000) and subsequent weakening (2000-2020) of the mean meridional circulation due to changes of the Rossby wave activity (both stationary and transient) over these two periods.

The stratospheric temperature trend in the NH DJF at high latitudes seen in the reanalyses (and transient experiments) could potentially be associated with influences other than external forcing due to ozone and $CO_2$, such as low-frequency modes of internal variability such as the Interdecadal Pacific Oscillation (IPO, (Meehl et al., 2016)), or forcing due to explosive volcanics. We argue here that our simulation design and the analysis performed here suggest that the low frequency climate variability and explosive volcanics are not responsible for the NH DJF stratospheric temperature behavior discussed here, but rather the dynamical response to the ozone and $CO_2$ changes.

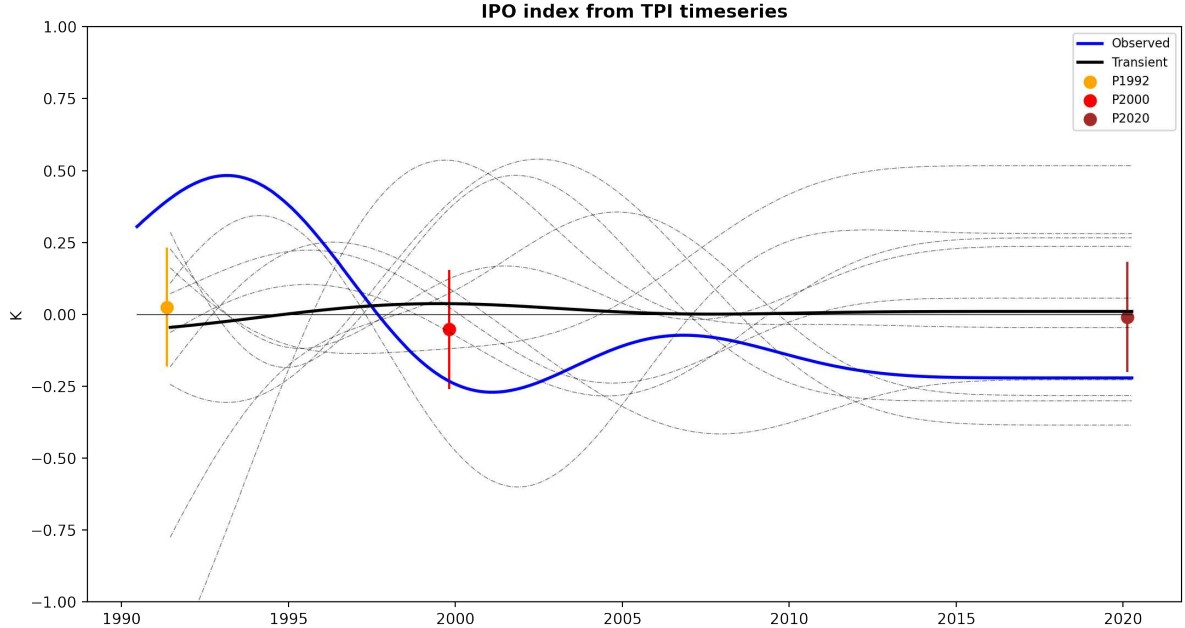

**Figure 7.** Tripole Index (TPI) for the Interdecadal Pacific Oscillation (IPO) timeseries is shown for observed data (blue line, NOAA OI SST V2) and model simulations. The black line shows the 10-member ensemble mean, while dotted lines show individual ensembles. Perpetual experiments are shown with their 1-std spread for P1992 (orange), P2000 (red), and P2020 (brown). Anomalies are calculated based on 1992-2020 baseline year.

250    We illustrate the dismissing of low frequency variability as responsible for the NH stratospheric polar temperature behavior with the IPO, as detected using the Tripole Index (TPI). The global mean surface temperature trend is positive during the positive phase of the IPO, and negative during the negative phase. The observed TPI is shown as the blue line in Figure 7. The transient experiment ensemble members TPI are shown in the dashed lines, and the transient experiment ensemble mean is shown with the solid black line. The influence of the IPO in the transient experiment ensemble mean is essentially removed,

255 so the behavior of the ensemble mean stratospheric temperature in the ensemble mean of the transient experiments cannot be due to the IPO. Similar arguments can be made about other low frequency modes of variability. In addition, the ensemble mean TPI from the perpetual experiments, shown by the yellow, orange and brown dots in the figure, also show a damped mean IPO signal. Experiments analyzed in detail here are a 30-member ensemble mean of 'perpetual year' experiments, so the contribution to our results from low-frequency modes such as IPO is negligible. The wide spread of IPO phases and the

260 damping of the IPO in the ensemble mean is due in part to our sampling of spun-up initial conditions, and due in part to the model's fidelity in capturing the propagation speed of the IPO. The inability of ensembles of free-running models to capture the IPO is consistent with the findings of Meehl et al. (2014), who determined that only a handful of CMIP ensembles (randomly)

whose IPO phases matched observations could successfully reproduce the observed global mean surface temperature trend (Meehl et al., 2014).

Dismissing the influence of volcanics on the temperature trends being discussed in this study is more straightforward. In our experiments the spun-up initial state's aerosol field does not contain aerosols emitted from Pinatubo (Supplementary Fig. S2), and explosive volcanic emissions are not included in the emissions that drive the interactive aerosol model, so the influence of stratospheric aerosol changes does not impact our results.

Our conclusions, therefore, are that the response to $CO_2$ and ODSs changes is robust and that the influence of internal variability or stratospheric aerosol can be neglected. Our results are consistent with the findings from Manzini et al. (2018), who found that the stratospheric polar vortex responds non-linearly to 2-K warming, contrasting the first and second 2-K warming periods over a transient 1% $CO_2$ ensemble of simulations. Manzini et al. (2018) found, due to only $CO_2$ change in the transient climate the polar vortex response is opposite and nonlinear. Abalos et al. (2019) reported a similar temperature response for the southern hemisphere during Austral summer and attributed it to changes in ODS and greenhouse gas concentrations. They, however, did not robustly identify a similar signal in the northern hemisphere.

Given the analysis here of the polar stratospheric response to external radiative forcing, an examination of the CMIP6 model output seems warranted. The CMIP6 multi-model mean NH DJF stratospheric temperature (65°N-90°N) shows no sign of warming from the year 1992 (Fig. 8a), and instead shows the gradual cooling that is expected based on theory. However, many CMIP models generally struggle to produce a realistic stratospheric circulation because of the low model top, insufficient vertical resolution, and inadequate aspect ratios between horizontal and vertical grids (Charlton-Perez et al., 2013; Hardiman et al., 2012; Rao and Garfinkel, 2021; Hall et al., 2021). We examined a single CMIP6 model with a well-resolved stratosphere, the NASA GISS E2 model's "Hi-Top" simulation (2-H) which is a part of the CMIP6 Historical experiment ensemble (Bauer et al., 2020; Kelley et al., 2020; Orbe et al., 2020) (Fig. 8b). Interestingly, the "Hi-Top" simulation does show a warming trend in the NH DJF stratospheric temperature in the first period (1992–2000) (7-year running mean). Given the noise from internal variability and different climate sensitivities to $CO_2$ and ozone forcing, the sharp warming after 1992 in the NASA GISS E-2-2H "Hi-Top" simulation is very similar to reanalyses and to our GEOS-MITgcm simulations, although the peak warming is not perfectly aligned (Figs. 1, 8b). In contrast, the separation between the two trend regimes is not present in the "Low-Top" simulation from the NASA GISS E2-1-H historical experiment (Fig. 8b). This examination suggests that modeling the proper response of the polar stratosphere to external forcing changes requires a well-resolved stratosphere to accurately capture the heat transport.

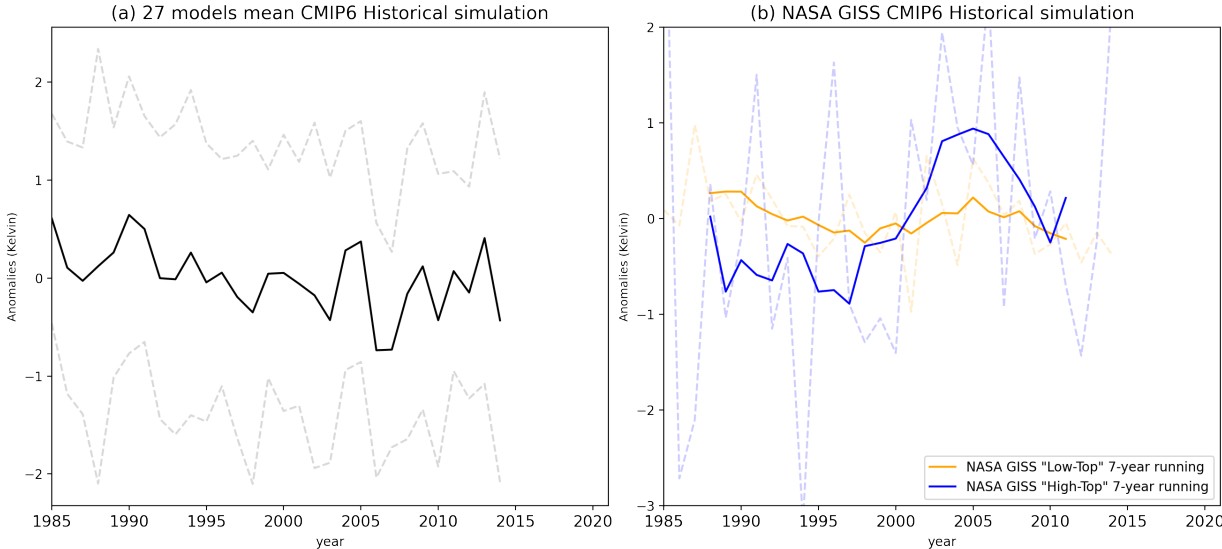

**Figure 8.** NH DJF mean 150 hPa-10 hPa and 65°N-90°N mean air temperature for (a) CMIP6 27 models mean Historical simulation (bold black line), and 1 std model spread are in dotted grey line. (b) shows NASA GISS "Low Top" (orange) & "High Top" (blue) simulation 7-year running mean, and actual anomalies are in dotted lines.

## 5    Summary

Examination of reanalyses (MERRA-2 and ERA5) shows that a cooling pattern in temperature trends has existed in the NH polar stratosphere during boreal winter from the 1980s to the 2020s (Fig. 1a), with the exception of a warming pattern from 1992 to 2000.

Our perpetual year 30-member ensemble mean GEOS-MITgcm experiments show that the NH DJF stratospheric polar temperature (65°N-90°N) response to the external forcing change (levels of $CO_2$ increase and ODSs change) in the atmosphere exhibits a general cooling with a warming phase in 1992-2000. The difference P2000-P1992 shows a temperature increase, whereas the temperature decreases in the P2020-P2000 (Fig. 3) difference. This opposing temperature response is in contrast with the general expectation that with increased $CO_2$ level in the atmosphere, the stratospheric temperature cools.

We further analyzed a 10-member ensemble mean of transient experiments from the year 1992-2020 forced with historical $CO_2$ levels and other observed external forcing. The results show a positive temperature trend during the year 1992-2000, and a negative temperature trend during the year 2000-2020 in the NH DJF polar stratosphere (Fig. 1a). These opposing temperature trends are consistent with the response of the perpetual experiments and reanalysis.

The longwave cooling in the NH DJF high-latitude stratosphere doesn't contribute to the local temperature response quali-305    tatively but rather responds to it. That is, the longwave decreased (cooling increased) with temperature increase in the P2000-P1992, whereas the longwave increased (cooling decreased) with the temperature decrease in the P2020-P2000 (Fig. 4).

We have shown that meridional heat transport resulting from $CO_2$ and ODS changes led to increased warming during the 1992–2000 period.

Our 30-member ensemble mean perpetual experiments and 10-member ensemble mean transient experiment starting from
a previously spun-up simulation have no realistic low-frequency variability, so the influence of the low-frequency variability (e.g. ENSO, IPO, PDO, & NAO) is dismissed as a driver of the behavior seen here. The aerosol content is also initialized from a spun-up initial condition, and the emissions supplied to the GOCART model do not contain explosive volcanics, so the stratospheric aerosol can't be the driver of the behavior shown in our simulations and in reanalyses. This suggests that only the level of $CO_2$ and ODSs change is driving the behavior pattern that we see in the period from the year 1992-2020.
The opposing polar stratospheric NH temperature trends in the two periods examined here are strongest during boreal winter. In the Southern Hemisphere, the polar stratospheric temperature also exhibits opposing behavior in these periods during austral winter and summer, most likely due to changes in the ODS. However, further study is needed to understand what is driving this phenomenon. Further study is needed to understand the seasonality and the regional climate response to the different levels of $CO_2$ forcing and ODSs separately.
We note that the end-to-end physical explanation of the wave activity in response to $CO_2$ and ozone changes that results in the cooling–warming–cooling behavior in the polar stratosphere is not shown here. We have, however, offered a physical interpretation of the proximate cause of this behavior. We have shown that the warming during 1992–2000 is related not to radiative effects, low-frequency modes, or volcanic influences, but to dynamical processes. We have traced the dynamical heating to the meridional eddy heat flux and the associated differences in wave activity. The remaining physical interpretation,
explaining why the wave activity changed during the warming period, remains the subject of ongoing studies. Other aspects of the modeled and observed response to external forcing changes will also be examined in future work based on our simulations. Studies like the one reported here are critical for the ability to predict the climate system on seasonal to decadal time scales.

**Appendix A**

The time-mean zonal mean meridional heat transport can be decomposed into zonal mean and zonal asymmetric components
as (following Peixóto and Oort (1984), eqn 4.9):

$$[\overline{VT}] = [\overline{V}][\overline{T}] + [\overline{V^* \, T^*}] + [\overline{V'T'}] \tag{A1}$$

Where [ ] shows the zonal mean and ∗ shows the zonal deviation of a variable. Here, $[\overline{V}][\overline{T}]$ represents the contributions of flux by steady symmetric circulations, $[\overline{V^* \, T^*}]$ represents flux contribution by stationary eddies, and $[\overline{V'T'}]$ represents contribution of co-variance of meridional wind anomaly and temperature anomaly (transient eddies) to the meridional heat
transport.

The TEM residual mass stream function, $\Psi^*$ is calculated as follows (Andrews et al., 1987; Birner and Bönisch, 2011)

$$\Psi^* = \Psi - ag^{-1}\cos\phi \frac{[v^*\theta^*]}{\partial_p[\theta]},\tag{A2}$$

where

$$\Psi = ag^{-1}\cos\phi \int_0^p [v]dp' \tag{A3}$$

The Eliassen Palm (EP) flux is defined as (Edmon Jr et al., 1980) :

$$\{F_\phi, F_p\} = \{-acos\phi[u^*v^*], facos\phi([v^*\theta^*]/[\theta_p]\} \tag{A4}$$

Where, $\phi$ is latitude and $p$ is pressure, $a$ is the radius of the Earth, $f$ is the Coriolis parameter, $\theta$ is potential temperature, $u$ is zonal wind, and $v$ is the meridional wind.

*Author contributions.*  A.F., A.M., and D.M. contributed to the design and analysis of this study. A.F. carried out simulations. A.T., E.S.,
K.W. contributed to the writing and experimental design. All authors contributed to the ideas, analysis, and writing of the manuscript.

*Competing interests.*  Authors have no competing interests as defined by Springer, or other interests that might be perceived to influence the results and/or discussion reported in this paper. All of the material is owned by the authors and/or no permissions are required. No conflict of interests.

*Acknowledgements.*  DM carried out research at the Jet Propulsion Laboratory, California Institute of Technology, under contract with NASA.
We would like to thank Clara Orbe (NASA GISS, NY, USA) for contributing to the initial discussion of the manuscript. High-end computing resources were provided by the NASA Center for Climate Simulate (NCCS) and the NASA Advanced Supercomputing (NAS) Division of the Ames Research Center.

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

**Table S 11.** CMIP6 Model List

| Model Name | Model Name | Model Name |
| --- | --- | --- |
| 1 AWI-CM-1-1-MR | 10 EC-Earth3-Veg | 19 CESM2-FV2 |
| 2 BCC-CSM2-MR | 11 IPSL-CM6A-LR | 20 CESM2-WACCM |
| 3 BCC-ESM1 | 12 MIROC-ES2L | 21 CESM2-WACCM-FV2 |
| 4 CAMS-CSM1-0 | 13 MIROC6 | 22 NorESM2-LM |
| 5 FGOALS-g3 | 14 HadGEM3-GC31-LL | 23 GFDL-CM4 |
| 6 CanESM5 | 15 UKESM1-0-LL | 24 GFDL-ESM4 |
| 7 CNRM-CM6-1 | 16 MRI-ESM2-0 | 25 NESM3 |
| 8 CNRM-ESM2-1 | 17 GISS-E2-1-G | 26 SAM0-UNICON |
| 9 E3SM-1-0 | 18 GISS-E2-1-H | 27 MCM-UA-1-0 |

**Supplementary information**

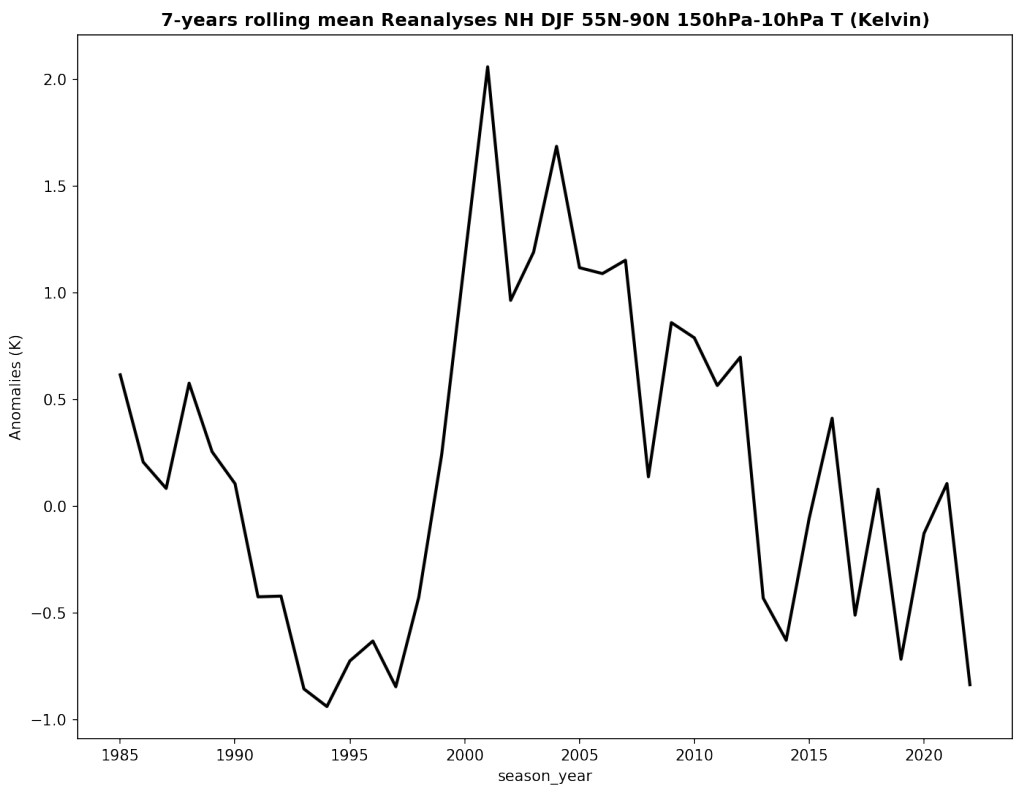

**Figure S1.** Rolling mean T from reanalysis

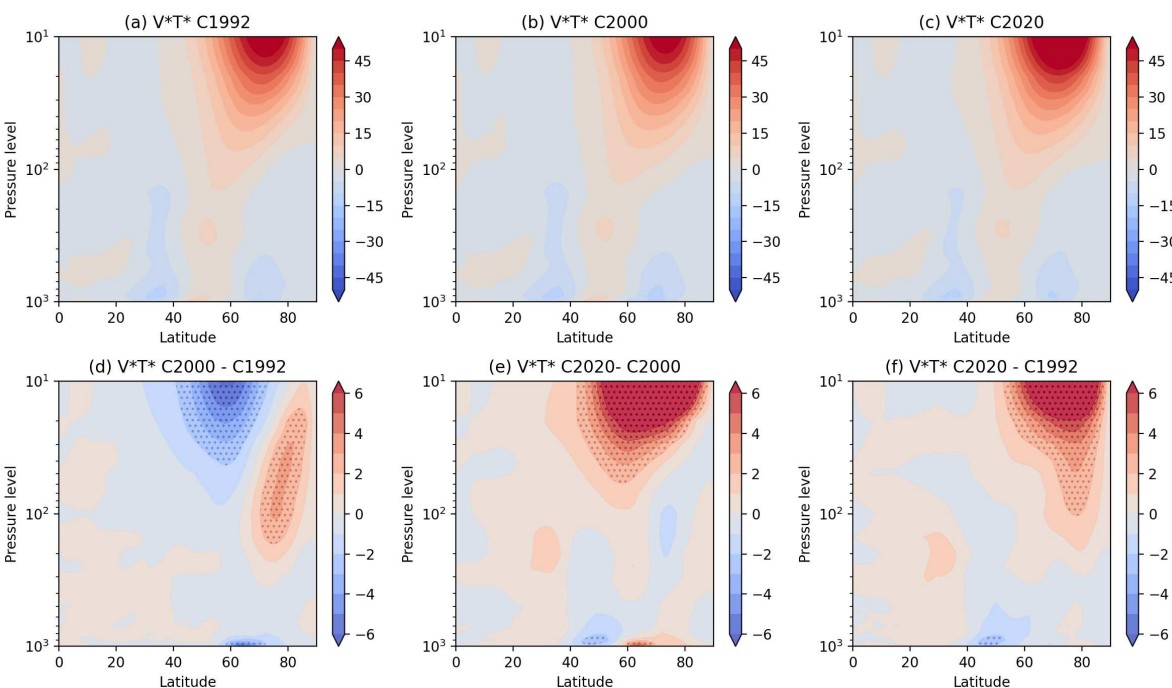

**Figure S2.** Zonal Mean NH DJF $V^*T^*$ mean for (a) P1992, (b) P2000, (c) P2020, and (d) difference between P2000 - P1992, (e) difference between P2020 - P2000, (f) difference between P2020 - P1992. Figures are stippled at 95% significance computed using a difference of means 2-sided t-test from 30-member ensemble sample.

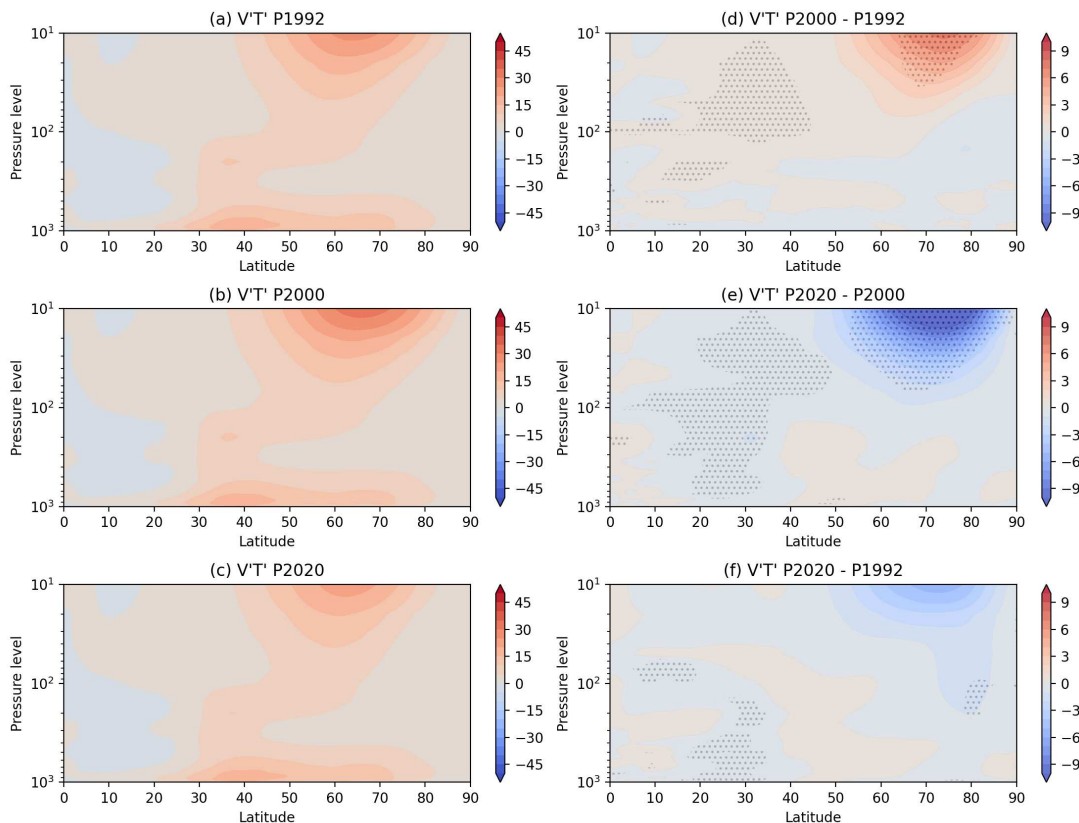

**Figure S3.** Zonal Mean NH DJF V'T' mean for (a) P1992, (b) P2000, (c) P2020, and (d) difference between P2000 - P1992, (e) difference between P2020 - P2000, (f) difference between P2020 - P1992. Figures are stippled at 95% significance computed using a difference of means 2-sided t-test from 30-member ensemble sample.

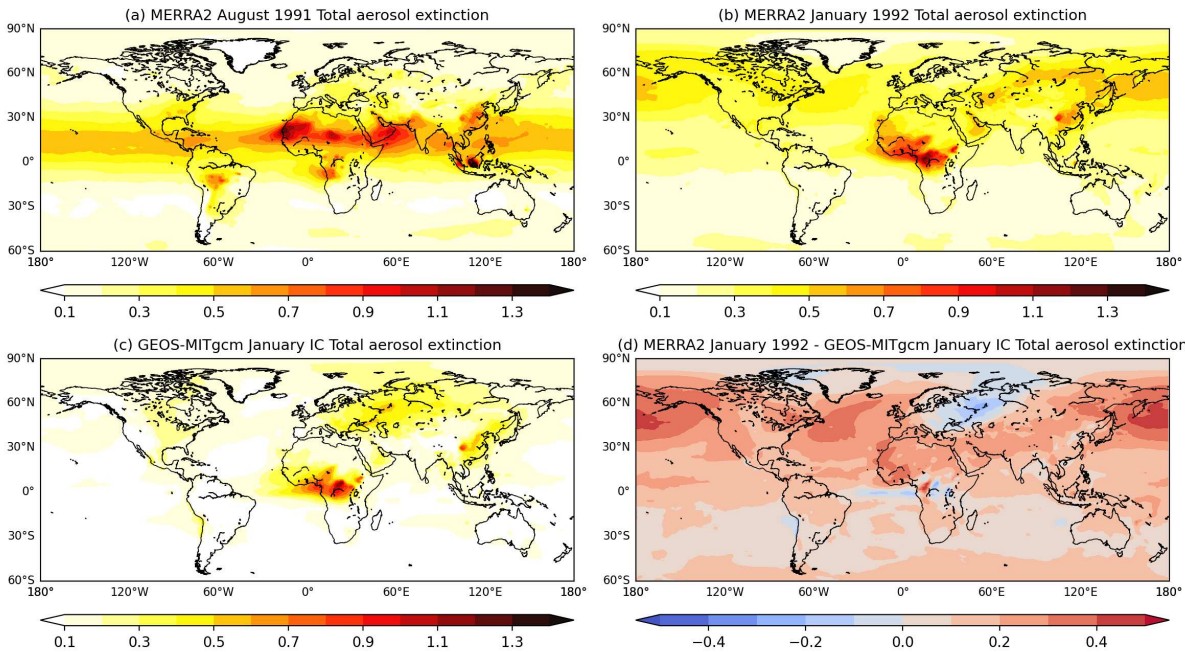

**Figure S4.** Total monthly mean aerosol extinction Aerosol Optical Depth [550 nm] for (a) MERRA2 August 1991, (b) MERRA2 January 1992, (c) GEOS-MITgcm January initial condition, which is used or more spun-up from this initial condition was used to initialize all experiments, and (d) MERRA2 January 1992 - GEOS-MITgcm January initial condition. The volcanic aerosol from Mount Pinatubo is visible in the MERRA2 August 1991 aerosol extinction in plot (a) (15°N, 120°E). The GEOS-MITgcm January initial condition is significantly different from observed state compared to the MERRA2 shown in plot (d), where any volcanic aerosol is not present.