# Peer review of "Northern Hemisphere Stratospheric Temperature Response to External Forcing in Decadal Climate Simulations"

_EGUsphere, 2025_

## Author Response (AR2)

**Northern Hemisphere Stratospheric Temperature Response to External Forcing in Decadal Climate Simulations**

Abdullah A. Fahad, Andrea Molod, Krzysztof Wargan, Dimitris Menemenlis, Patrick Heimbach, Atanas Trayanov, Ehud Strobach, and Lawrence Coy

**Reviewers' comments are in black, responses are in blue color**

**Revised manuscript link:**

https://www.researchgate.net/publication/362231098_Northern_Hemisphere_Stratospheric_Temperature_Response_to_External_Forcing_in_Decadal_Climate_Simulations

**A tracked change version of the manuscript is attached.**

**RC1:** The study by Fahad et al. aims to explore the stratospheric temperature response to external forcing. To this end the authors analyze a series of 30-year perpetual time slice experiments with forcings corresponding to year 1992, 2000, and 2020 conditions. The authors motivate their study with a stratospheric temperature increase during 1992-2000 and decrease during 2000-2020. While I appreciate the authors aim to advance our understanding of stratospheric temperature trends and variability, I identify a series of shortcomings in the presented work.

We thank the reviewer for their insightful suggestions to improve the manuscript. We have revised the manuscript in accordance with the reviewer's comments, detailed below.

Specific Comments:

1) The time periods for trend analysis: the authors motivate their analysis with opposing trends between 1992-1999 and 2000-2020. I consider the first time period too short for a robust trend analysis. Similar positive sloping trends could be randomly identified e.g. at the end of the time series in Fig. 1b. Overall Fig. 1 highlights strong inter-annual variability in the temperature evolution over the Arctic polar cap. To provide further insights into the drivers of this variability – and its peak amplitudes – would be a worthy endeavor and I suggest the authors to focus their analysis on this rather than short-term trends or tendencies.

We agree that focusing on the 1992–2000 period for trend analysis may appear arbitrary given the large interannual variability. To address this, we have computed a 7-year running mean of polar stratospheric temperature over the entire 1981–2024 reanalysis period, which filters out much of the interannual variability. The results are shown in our new Figure 1. The longer,

filtered time series reveals a clear cooling trend in the polar stratosphere throughout the period, except during 1992–2000, when temperatures increase dramatically. From this plot, it is evident that the end of the initial cooling period and the onset of warming occur in 1992, the peak of the transient warming is reached in 2000, and a renewed cooling trend begins after 2000 to 2020s. We believe that this filtered time series clarifies the rationale for selecting our analysis periods (1992–2000 and 2000–2020). Figure 1 has been updated accordingly, and the manuscript text has been revised to explain these changes more clearly.

We have updated the following in the manuscript:

*Figure number: 1, supplementary figure S1*
*Text Line numbers: lines 2-7, lines 77-85, lines 166-173.*

2) Simulation design and ensemble size: the authors investigate 30-year perpetual simulations with 1992, 2000, and 2020 conditions. Each year of these simulations is treated as single ensemble member that are pooled for composite analysis. Generally, I consider this ensemble size too small to derive robust conclusions especially, given the large inter-annual variability in NH polar cap stratospheric temperature. Commonly single forcing studies focusing on stratospheric temperature effects have utilized 100 year (+) plus integrations. How robust are the findings against sub-sampling? How different/overlapping are the polar cap temperature distributions across these 30-year sets? How would the results change if a bootstrap analysis is applied?

We agree that a 30-member ensemble size can be small for understanding some aspects of the stratospheric response under climate change; however, our 30-member "perpetual" ensemble plus a 10-member "transient" ensemble shows robust agreement of NH polar vortex warming from 1992 to 2000 and cooling from 2000 to 2020. We used a t-test on the 30-member ensemble and applied stippling to indicate 95 % significance for the mean differences (all figures in the manuscript that include mean-difference panels use this convention). Additionally, as suggested, we performed a nonparametric bootstrap significance test using $\alpha = 0.05$ (95 % confidence) and 1,000 bootstrap samples to assess whether the mean difference between the two datasets is statistically significant without assuming normality. We still found the NH winter polar vortex temperature increase between 1992 and 2000 to be significant. Next, we randomly subsampled the 10-member ensemble five times and applied the same bootstrap test; even the least significant subsample shows warming from 1992 to 2000 at 95 % confidence. Figures are shown below, and we have added this description to the manuscript text *(Line numbers: 159-163).*

[Figure]

**R.Fig. 1:** Same as Figure 3 in the manuscript, except stipplings are based on the bootstrapping method at a 95% significance level.

[Figure]

**R.Fig. 2:** Same as Figure 3 in the manuscript, except based on randomly sub-sampled (10-member) and the stipplings are based on the bootstrapping method at a 95% significance level.

3) How robust are the findings against the starting years selected?

We have analyzed starting years 1992, 1993, 1994, and 1995, and the conclusions from the 10‑member transient experiments remain unchanged: the polar stratosphere warms significantly during that period (95% confidence, Student's t-test). Figure 1 (updated) in the manuscript shows the mean trend of the transient experiments along with their spread, where the observed warming trend and model trend remain similar for different starting years nearby.

4) How does the vortex state and stability in these integrations compare with the observational record? And how different is it within and across the ensembles?

The following text and a new Figure 2 are added to the updated manuscript to show the vortex state comparing reanalysis to the GEOS-MITgcm simulations, demonstrating the fidelity of the simulations. *(Line numbers: 133-142)*

*"The GEOS-MITgcm coupled model transient simulations reproduce the mean state and variability of the polar vortex reasonably well as compared to reanalysis. The geopotential height at 10 hPa from the 10-member ensemble mean for January over 1992–2020 shows a similar mean state and location of the NH polar vortex during winter compared to reanalysis (MERRA-2: 1992–2020) (Fig. 2a,b). The 30-year mean January momentum‑flux variance (U'V') associated with the vortex wind jet, calculated from sub‑monthly fields, further shows that the stability and variability of the vortex core are simulated reasonably well in the GEOS‑MITgcm compared to the reanalysis (Fig. \ref{vortex} c,d). The model, however, produces a bit weaker mean geopotential height (~ 4km higher at the core) and momentum flux variance (weaker ~ 25 m2/s2 near Greenland) compared to MERRA-2, suggesting that the simulated vortex may be a bit more resilient under extreme wave forcing. This could be a result of a lack of higher vertical resolution in the stratosphere. However, the close agreement in both mean height structure and eddy flux confirms that the model faithfully represents the polar vortex's stability and natural variability."*

5) In analogy to comments 2+4), how different are the EHF (and stationary and transient terms) within these integrations? How robust are the findings against bootstrapping?

Figure 5, Figure 6, and supplementary Figure S1 show the mean differences in eddy heat fluxes. The figures are stippled to indicate 95 % significance, computed using a two-sided difference-of-means t-test on the 30-member ensemble. The bootstrapping method, with 1 000 iterations at the 95 % confidence level, yields similar results.

6) The study is based on ensembles obtained with a single model, which does not particularly well align in terms of variability (trends and their significance) with reanalysis data (as shown in Fig. 1). I would suggest including in a revised manuscript additional models to corroborate the results.

We agree that the previous Figure 1 was somewhat confusing for interpreting the variability and trends between the reanalysis transient years and the 10-member ensemble-mean transient experiment. We have replaced it with a new Figure 1, in which a 7-year running mean, applied to minimize internal variability, shows behavior very similar to that of the 10-member ensemble-mean polar-vortex temperature. This result holds for both smaller and larger running-mean windows, although variability increases with smaller windows and is further smoothed with larger ones. A comparison with similar perpetual experimental designs is beyond the scope of this study but will be explored in future work. For the transient experiment, we have examined both high-top and low-top historical CMIP simulations and discussed the results in the manuscript (Fig. 9).

7) The authors refer several times to supplementary figures, which I could unfortunately not find enclosed in the preprint or linked to a research square.

Uploaded a single PDF with supplementary figures attached at the end.

**Northern Hemisphere Stratospheric Temperature Response to External Forcing in Decadal Climate Simulations**

Abdullah A. Fahad, Andrea Molod, Krzysztof Wargan, Dimitris Menemenlis, Patrick Heimbach, Atanas Trayanov, Ehud Strobach, and Lawrence Coy

**Reviewers' comments are in black, responses are in blue color**

**Revised manuscript link:**

https://www.researchgate.net/publication/362231098_Northern_Hemisphere_Stratospheric_Temperature_Response_to_External_Forcing_in_Decadal_Climate_Simulations

**A tracked change version of the manuscript is also attached.**

**RC2:** The paper investigates the recent decadal variations of the polar stratospheric temperature during boreal winter (DJF) by analyzing the last 30 years of reanalysis datasets, simulations of a 1-degree ocean-atmosphere coupled GCM (GEOS-MITgcm) and CMIP6 historical simulations. Two types of simulations are performed with the coupled GCM, some are forced with time-evolving CO2, ozone and aerosols from 1992 to 2020 and called transient simulations. Some others have constant concentrations of CO2, ozone and aerosols related to a given year and called "perpetual year" simulations. The authors detect a positive trend in polar stratospheric temperature from 1992 to 2000 and negative trend from 2000 to 2020 in both the ensemble mean of the transient simulations and reanalysis. They also show that the "2000 perpetual year" simulations have higher stratospheric temperature than the "1992" and "2020 perpetual year" simulations. The setup of the "perpetual year" simulations is interesting to investigate the stratospheric temperature changes during the last decades.

However, no physical interpretation of the results is provided. The heat fluxes budget show that wave propagation and breaking does play a role. But there is no interpretation of why waves behave as they do for the different experiments. So the paper does not provide any direction of why 1992, 2000, 2020 generated different behaviors in the waves.

We agree with the reviewer that we have not provided a definitive end-to-end physical explanation of the response to $CO_2$ and ozone forcing that results in the cooling–warming–cooling behavior in the polar stratosphere. We have, however, offered a robust physical interpretation of the proximate cause of this behavior. We have shown that the warming during 1992–2000 is related not to radiative effects, low-frequency modes, or direct volcanic emission influences, but to dynamical processes. We have traced the dynamical heating to the meridional eddy heat flux and the associated differences in wave activity. We have further

suggested the plausible explanation that radiative heating changes in the tropics due to ozone, potentially related to the Pinatubo volcanic emissions,    resulted in the stronger TEM and the increased meridional heat transport during the 1992-2000 period. We have explained further with a new figure and texts:

*Figure number: Figs. 5, 6, and 8*
*Text Line numbers: 264-284*

We have added the following caveats and texts in the manuscript in response to these questions. *(Line 322-329)*

I have another major concern related to the detected trends. Computation of trends is quite strange when analyzing 8 consecutive years from 1992 to 2000. I think these trends for this small period are not significant (in particular ERA5 in Fig.1b or high-top simulations in Fig.8b).

We agree that focusing solely on the 1992–2000 period for trend analysis can be seen as arbitrary. However, examining the centered 7-year running mean of polar-stratosphere temperature over the entire 1981–2024 reanalysis period, thereby minimizing interannual variability, reveals a clear cooling trend in the polar stratosphere, except during the 1992–2000 interval, when temperatures increase dramatically. This is the rationale for examining the warming trend during 1992-2000. We selected 1992, 2000, and 2020 to investigate and compare the end of the initial cooling period (1992), the peak of the transient warming (2000), and the recent resumption of cooling at the end of the time series (2020). We have updated Figure 1 to reflect this rationale and revised the manuscript text to explain these changes more clearly.

We have updated the following in the manuscript:

*Figure number: 1, 9,  supplementary figure S1*
*Text Line numbers: lines 2-7, lines 77-85, lines 166-173.*

Also from a statistical significance point of view, it would be important to show the spread of the 30 members in Fig.1a to see if the ensemble means are significantly different.

We have shown the spread of the ensemble with a shaded line in the updated fig. 1. We used a t-test on the 30-member ensemble and applied stippling to indicate 95 % significance for the mean differences (all figures in the manuscript that include mean-difference panels use this convention). Additionally, as suggested, we performed a nonparametric bootstrap significance test using $\alpha = 0.05$ (95 % confidence) and 1,000 bootstrap samples to assess whether the mean difference between the two datasets is statistically significant without assuming normality. We still found the NH winter polar vortex temperature increase between 1992 and 2000 to be significant. Next, we randomly subsampled the 10-member ensemble five times and applied the

same bootstrap test; even the least significant subsample shows warming from 1992 to 2000 at 95 % confidence. Figures are shown below, and we have added this description to the manuscript text. *(Line numbers: 159-163).*

[Figure]

**R.Fig. 1:** Same as Figure 3 in the manuscript, except stipplings are based on the bootstrapping method at a 95% significance level.

[Figure]

**R.Fig. 2:** Same as Figure 3 in the manuscript, except based on randomly sub-sampled (10-member) and the stipplings are based on the bootstrapping method at a 95% significance level.

To conclude, even though the perpetual year simulations are interesting, the paper does not provide any interpretation of the different behaviors of the three years 1992, 2000, 2020 and there is a strong lack of significance tests. How do the various changes in CO2, ozone and aerosols influence the waves and residual mean circulation ? Such a question should be adressed in the paper but this is not the case. Therefore I recommend rejection of the paper even though I must admit there is some potential for publication in the future but with an entirely revisited

paper including deeper analysis of the simulations and an interpretation of CO2/ozone/aersols effects.

We hope that we have addressed these questions already in our responses to your previous comments above.

Major concerns:

I) About detected trends. In fig1.b, interannual variability is very large compared to the trends. The results of the detected trends for the first period (1992-2000) might strongly change if the year 2000 is included or not. Same thing in Fig.8b, by removing or adding a year for the high-top simulations, the sign of the detected trends might change.

We agree that focusing solely on the 1992–2000 period for trend analysis can be seen as arbitrary. However, examining the centered 7-year running mean of polar-stratosphere temperature over the entire 1981–2024 reanalysis period, thereby minimizing interannual variability, reveals a clear cooling trend in the polar stratosphere, except during the 1992–2000 interval, when temperatures increase dramatically. This is the rationale for examining the warming trend during 1992-2000. We selected 1992, 2000, and 2020 to investigate and compare the end of the initial cooling period (1992), the peak of the transient warming (2000), and the recent resumption of cooling at the end of the time series (2020). We have updated Figure 1 to reflect this rationale and revised the manuscript text to explain these changes more clearly.

We have updated the following in the manuscript:

*Figure number: 1, 9, supplementary figure S1*
*Text Line numbers: lines 2-7, lines 77-85, lines 166-173.*

II) Significance tests. In addition to I) about trends, we do not know if the difference between the 30-year ensemble means are significantly different. Please show all the members, or least add the max and min among all members.

We have performed a two‑sided t‑test to assess the significance level. All figures showing the difference in means are stippled at the 95% significance level, using 30‑member ensembles as samples. In addition, as suggested, we have conducted bootstrapping and sub‑sampling significance tests, which are more stringent and still show consistent, statistically significant warming.

We have updated Figures 1, 7, and 9 to show the ensemble spread for both transient and perpetual experiments.

III) Section 3.2 is entitled "dynamical mechanism of forced change" but I do not see any interpretation of why the waves could change their propagation and breaking as function of the different forcings. So in my opinion, there is no proposed mechanism to explain the observed changes in temperature and waves propagation. All the figures from 2 to 7 are consistent with each other but this only provides a part of potential mechanims to explain the changes.

To assess how wave activity affects meridional heat transport, we analyzed tropical stratospheric radiative heating and associated ozone changes.

In response, we added a new figure (Fig. 8) and accompanying text (lines 264–284).

IV) Simulations. The authors mention several times that the changes cannot result from low-frequency modes of interannual variability but they do not show any evidence. Line 102, the initial states are said to be uncorrelated but which variables have been looked at ? SSTs ? Also it is not clear to me if the 30 initial states are the same for the P1992, P2000 and P2020 simulations.

A new Fig. 7 shows that both the 10-member ensemble simulations and the perpetual experiments have initial states that differ from the observed IPO index phase. Although the ensemble mean (bold thick line in Fig. 7 minimizes the variability of the IPO index, we still observe a statistically significant polar vortex warming trend in our experiments. The large spread of IPO phases also reflects that our initial conditions were randomly chosen from previously spun-up simulations.

We have added text on choosing initial conditions and influence of low frequency modes on text: line numbers: 249-263

V) Abstract. I found the abstract not well written. It should first mention decadal variations in reanalysis and results of the transient simulations before desribing the "perpetual year" simulations. Analysis of heat fluxes cannot be called "analysis of the temperature budget (line 13) since a full temperature budget would require to show the diabatic terms too.

We have updated the abstract as suggested. Our temperature budget terms included all terms, including diabatic terms. However, only significant term changes are discussed in the manuscript (mentioned in the text: Line 13).

VI) Wording. In some sentences, the text contains too strong statement. For instance, line 249 "high-top simulation" is said to be "very similar to reanalysis and GEOS-MITgcm simulations". I do not think the trends of the first period exist in reanalysis (Fig.1b) or high-top simulation (blue curve in Fig.8b) and that the curves resemble to each other.

We have updated the figures and text to reflect where is the similarities between the CMIP GISS run and GEOS-MITgcm and reanalysis.

Another example is in line 283-284 "The opposing polar stratospheric NH temperature trends in the two periods examined here are only significant during boreal winter". Why is the word "significant" used here ? In Fig.1 I do not see any significance tests.

Text updated

VII) Supplementary information. I was not able to download it from the web sites.

Uploaded a single PDF with supplementary figures attached at the end of the manuscript.

---

## Author Response (AR3)

**Northern Hemisphere Stratospheric Temperature Response to External Forcing in Decadal Climate Simulations**

Abdullah A. Fahad, Andrea Molod, Krzysztof Wargan, Dimitris Menemenlis, Patrick Heimbach, Atanas Trayanov, Ehud Strobach, and Lawrence Coy

**Reviewers' comments are in black, responses are in blue color**

I had two major concerns in my previous review. One was about the significance of the trends. Rather than computing trends over a few years which does not make sense in my opinion the authors now shows the spread of the ensemble of transient and perpetual-year simulations in Fig.1b. This gives a better insight of the involved uncertainties. I am now more convinced by the first figure and the fact that P2000 and P2020 simulations are significantly different from the P1992 simulation. Figure 8 has also been modified in a similar manner and I found it more satisfying.

My second major concern was about the lack of dynamical interpretation of the change in wave dynamics between P1992, P2000 and P2020 simulations. The authors argue that the paper provides already enough material by discarding the effect of low-frequency variability, volcanoes, radiation and the reasons for the change in wave dynamics is left for future studies. I understand the authors' view but it is a bit strange to finish a paper without providing any hypotheses or future directions of research to look at this problem.

The reader stays quite frustrated at the end and it would be nice to provide some sentences on how to tackle this problem in the future**.** At the same time, the paper has been reinforced by inserting a new figure (Figure 7) showing there is indeed no systematic low-frequency variability that could explain the differences between the different experiments.

I found this additional figure useful and helps convincing the reader. Despite several important improvements, there are several aspects of the paper that appear unclear to me and I would recommend publication once the authors carefully answered the following comments:

We appreciate the reviewer's positive feedback regarding the revisions. We have addressed the remaining concerns regarding the dynamical interpretation and the specific points of clarification individually below.

**Major comments:**

- Figure 8: I do not understand how it is possible that a high-top CMIP6 model performs less well than the 1°GEOS-MITgcm coupled model. Indeed, the GEOS-MITgcm model reproduces the timing of the warming between 1992 and 2000 very well (Figure 1b) where as the NASA GISS "high top" has a clear delay in the warming (Figure 8). The authors argue about the importance of resolving the stratosphere correctly at the end of the paper but what is the vertical resolution of the GEOS-MITgcm ? Is it low-top or high-top ? The model having 72 levels I would guess that its vertical resolution of the stratosphere is coarser than that of the NASA GISS hightop model.

The initial conditions for the ensemble simulations with GEOS-MITgcm are relatively recent in relation to the stratospheric warming period, and may experience some advantage in the timing of the stratospheric warming relative to CMIP6 historical simulations. The figure and the discussion in the text show the total lack of a simulated warming at any time in the low top simulation, due, in part, we suggest, to the lack of resolution in the model's stratosphere. We have added a line in the text describing Figure 8 to point out the difference between CMIP6 initial simulation dates and those of the GEOS/MITgcm simulations presented here.

Updated text Line numbers: 299-310.

- Figure 7: it is still not clear the reasons why the numerical protocol suppresses any effect of low-frequency modes. This should be better explained in the methodology section (see my minor comments). Is it related to the fact that initial conditions for the P1992, P2000 and P2020 simulations are the same ?

Figure 7 illustrates that the ensemble mean of 30 ensemble members effectively minimizes any low-frequency variability, despite the presence of low frequency variations in individual ensemble members. The first year of the P1992, P2000, and P2020 perpetual experiments were initialized from the same state  and were run freely with the forcings specific to their respective years. The presence of the low frequency modes in different phases in the different ensemble members (phase spread in low frequency modes), as shown in the figure, acts to negate the impact of the low frequency modes in the ensemble mean, thus removing low frequency variability as an explanation for the stratospheric warming event as evidenced by the different behavior in the perpetual simulations (P1992, P2000 and P2020).

Texts are added for both the perpetual and transient experiments in the m,anuscript to explain this issue further in the methodology section: Lines: 131-140

- About aerosols effect. The abstract says "Each simulated year of these perpetual experiments is forced with the CO2, Ozone, anthropogenic aerosol emissions" but at the end the conclusion is that the changes are said to be solely due CO2 and Ozone. How did the authors conclude that anthropogenic aerosol emissions do not play a role in their simulations ? Maybe I missed a key explanation in the paper.

The budget term for the aerosol radiative impact was evaluated in relation to the other budget terms, as discussed at the end of Section 3.1 and was deemed too small an impact on the temperature to play a role in explaining the temperature trends which are the main subject of the study. Lines: 211-217.

**Minor comments:**

- Caption Figure 1: "The blue line in (a) shows a 7-year running mean" to be replaced by "The DJF mean is shown in thin black line and its 7-year running mean in blue."

Updated

- Line 78: Was the sharp warming period from 1992 to 2000 reported in previous studies ? Please cite references if appropriate.

Only a limited number of studies have specifically examined the pronounced increase in lower stratospheric DJF Northern Hemisphere temperatures during 1992–2000. Nonetheless, related literature does exist, particularly investigations focusing on the effects of volcanic aerosol loading and the dynamics associated with sudden stratospheric warmings. We have cited relevant references addressing this topic in lines: 49-76.

- Line 117: I do not understand the following sentence "The 30 years of simulation are regarded here as a 30-member ensemble of simulations of the 'perpetual' year, as the initial states for each perpetual year are random."

We have added the following text "The external boundary conditions of the `perpetual' experiments repeat annually while the internal atmospheric state varies, and so each year of the simulation functions as an independent ensemble member (or realization) of that specific year's climate". Lines: 119-121

- Line 118: The authors say "The 30-member ensemble mean of these experiment does not include a realistic simulation of the phase of low-frequency modes of internal variability". This

statement should be carefully explained. Is it because the initial conditions of the P1992, P2000, P2020 simulations are the same ?

We ask the reviewer to see our response to the Major comment related to Figure 7, which raises the same question.

- Line 127: It would be better to start a new paragraph when starting the description of the transient experiments

Updated.

- Line 130: It would be good to explain why the numerical setup leads to the following statement: "The low-frequency SST modes are out of phase across these ensemble members"

Please see the response to the major comment about low frequency modes for a clarification of this statement.

- Line 135: please remove "variance". Could you tell the reader for which purposes momentum fluxes are plotted here.

We would like to retain the use of "variance," as the plots shown in Fig. 2c,d are the mean of $U'V'$, which represents the mean of the 6-hourly variance of the momentum flux associated with the vortex wind jet. This illustrates how the variability and stability of the vortex in our model simulations compare to the variability of the observed state as estimated from reanalysis. This analysis was added in response to the previous review, and we would like to retain it because it demonstrates how our model simulates the vortex itself, as explained in lines 141–151.

- Line 136: "calculated from sub-monthly fields" is not clear enough. What does that mean ? a subtraction to the monthly mean ?

Model output from our simulations included 6-hourly prognostic fields - these variances are computed as the deviation of the 6-hourly values from the monthly mean, and so we refer to these variances as "computed from sub-monthly fields". Text updated: 145-146.

- Line 147: Is the definition of the primes the same as in Line 135-136 ?

Yes

- Line 148: "overbar is the time average (DJF seasonal mean)". Is the DJF seasonal mean a climatological mean over all DJF months of all the years or is it dependent on the year ?

The overbar is the mean over all the years. We have updated the text.

- Line 156: please suppress "of"

updated

- Line 160: how do you select the samples ?

These are random samples, and we have updated the text to mention it.

- Lines 190-192: It would be good to refer to the dots in Fig.1b related to the results of the perpetual year experiments.

We have added text to mention the consistency between these two sets of experiments. Line number: 204-205.

- Line 194: "expected radiative effects of increasing CO2" please add a reference

Updated.

- Line 214: "three sets of simulations" to be changed by "three sets of perpetual-year simulations."

Updated.

- Lines 215-216: please replace "result" by "simulation"

Regarding the comment for Lines 215–216: We have checked the text and the word 'result' does not appear in these lines. The sentence reads "To articulate the role of the dynamic tendency terms on the NH stratospheric polar temperature we therefore begin by analyzing the eddy heat flux from the three sets of simulations. "

- Line 247: Please explain how the simulation design helps to discard the effect of low-frequency climate variability"

Please see our response to major comments on the same topic.

- Figure 7, caption: "dotted lines show individual ensembles" to be replaced by "dotted lines show individual simulations"

Updated

- Line 260-261: The end of the sentence is not clear. What is meant by "propagation speed of the IPO" ?

We have updated the text to say "correct phase" instead of "propagation speed".

- Line 287: as mentioned above, it is strange that GEOS-MITgcm performs better than the NASA GISS high top model because I think GEOS-MITgcm has a less well resolved stratosphere.

Please see our response to major comments on the same topic.

- Line 319: I am not sure to understand if ODS is the forcing in the simulations presented in the paper or if the model is forced by ozone concentrations

As described in the methods section, only Ozone concentrations were varied for the respective years.

- Line 325: Some hypotheses could be however provided. It is a bit weird to finish a paper without offering possible mechanisms or approaches to tackle that question.

As the revised text in the summary section now states more clearly, we have indeed proposed a mechanism that could explain the warming then cooling behavior seen in reanalyses, our model simulations and the GISS CMIP6 high-top simulation. We suggest a possible indirect impact of the Pinatubo eruption, whereby the emissions from the eruption had an impact on ozone chemistry and locally increased ozone concentrations. This additional ozone generated additional tropical heating and the subsequent increase in eddy heat transport to the polar stratosphere. The summary has been rephrased to make this suggested mechanism more noticeable. Line numbers: 336-344.